# High stress twinning in a compositionally complex steel of very high stacking fault energy

Zhangwei Wang [1✉], Wenjun Lu [2✉], Fengchao An[2], Min Song [1], Dirk Ponge[3], Dierk Raabe [3] & Zhiming Li [1,3,4✉]

Deformation twinning is rarely found in bulk face-centered cubic (FCC) alloys with very high stacking fault energy (SFE) under standard loading conditions. Here, based on results from bulk quasi-static tensile experiments, we report deformation twinning in a micrometer grain-sized compositionally complex steel (CCS) with a very high SFE of ~79 mJ/m$^2$, far above the SFE regime for twinning (<~50 mJ/m$^2$) reported for FCC steels. The dual-nanoprecipitation, enabled by the compositional degrees of freedom, contributes to an ultrahigh true tensile stress up to 1.9 GPa in our CCS. The strengthening effect enhances the flow stress to reach the high critical value for the onset of mechanical twinning. The formation of nanotwins in turn enables further strain hardening and toughening mechanisms that enhance the mechanical performance. The high stress twinning effect introduces a so far untapped strengthening and toughening mechanism, for enabling the design of high SFEs alloys with improved mechanical properties.

[1] State Key Laboratory of Powder Metallurgy, Central South University, 410083 Changsha, China. [2] Department of Mechanical and Energy Engineering, Southern University of Science and Technology, 518055 Shenzhen, China. [3] Max-Planck-Institut für Eisenforschung, Max-Planck-Str. 1, 40237 Düsseldorf, Germany. [4] School of Materials Science and Engineering, Central South University, 410083 Changsha, China. ✉email: z.wang@csu.edu.cn; luwj@sustech.edu.cn; lizhiming@csu.edu.cn

The plastic deformation mechanisms that govern the mechanical performance of crystalline metallic materials include dislocations, twinning, stacking faults, and displacive phase transformations[1]. While the motion of the former defects, i.e. of the dislocations, maintains lattice coherency, the latter three mechanisms create symmetry breaks, manifested by changes in the stacking sequence of the densely packed atomic planes. This crystalline defect is called stacking fault and its associated energetic penalty is referred to as the stacking fault energy (SFE)[2]. Kinematically, twins, stacking faults, and displacive phase transformations are carried by partial dislocations[3]. These have a smaller self-energy than complete lattice dislocations, but when activated, partial dislocations shift the lattice locally into the wrong configuration, thereby creating the stacking fault. For this reason, twins, stacking faults and displacive phase transformations, which can endow metals with excellent strain hardening features, are typically absent in bulk materials with relatively high SFEs, such as pure Al (166 mJ/m²) and Ni (125 mJ/m²)[2,4,5], in which the competing dislocation slip is energetically less costly[4,6]. Thus, except for some extreme cases[7–12], such as deformation of nanocrystalline Al films under microindentation[7] or bulk Al exposed to large strain rates[11], the deformation behavior of high SFE materials is governed by dislocations. As a result, bulk alloys with high SFE have till today not unleashed the excellent strain hardening reserves provided by mechanical twins and stacking faults.

According to previous studies conducted over the past decades, deformation twinning has also not been found in tensile loaded Fe–Mn–Al–C steels with high SFE (the upper limit for twinning is ~50 mJ/m²), a promising material class for highly demanding engineering applications due to their low mass density, excellent mechanical properties, and low costs[13]. The deformation of Fe–Mn–Al–C lightweight steels is initially dominated by planar slip of dislocations, which further evolves into slip bands consisting of high densities of dislocations as deformation proceeds[14]. Higher strength-ductility regimes remain inaccessible though for these alloys as the available strain-hardening mechanisms remain confined to dislocations and their interactions with grain boundaries and precipitates[15–17]. The twinning-induced plasticity (TWIP) effect, as a highly effective strain hardening and toughening mechanism enabling attractive mechanical properties[18,19], has remained inaccessible for these materials, due to their high SFEs.

Here, we report deformation twinning and the associated high strengthening effect in a lightweight compositionally complex steel (CCS) with a SFE of ~79 mJ/m². As noted above, deformation twinning would be usually expected to be impossible to occur in bulk materials with such high SFE under quasi-static tensile loading conditions. The CCSs are a class of materials developed by applying the concept of high-entropy alloys (HEAs) to the redesign of conventional lightweight Fe–Mn–Al–C steels[20]. The compositional degrees of freedom introduced by the high entropy concept allow to shift the material's overall composition into regimes where the formation of a unique blend of dual-nanoprecipitation of κ-carbides (ordered face-centered cubic, FCC) and B2 (ordered body-centered cubic, BCC) phases becomes possible, producing the high strength required to activate mechanical twins.

## Results
### Microstructures of the CCS
The CCS studied in this work has a nominal composition Fe-26Mn-16Al-5Ni-5C (at. %). After homogenization at 1200 °C, the cast steels were hot-rolled to 75 % total thickness reduction, followed by cold-rolling to 60% reduction in thickness and subsequent annealing at 900 °C for 3 min (see "Methods").

Figure 1 shows the microstructures of the material across several length scales. The alloy exhibits a partially recrystallized structure with B2 phase in an austenitic γ (FCC) matrix, revealed by the inverse pole figure (IPF) (Fig. 1a) and phase map (Fig. 1b) obtained from electron backscatter diffraction (EBSD). According to multiple EBSD maps, the recrystallized regions in the microstructure occupy a fraction of 57% with an average grain size of ~1.5 μm, while the rest (43%) are non-recrystallized regions with a larger average size of ~10 μm. The bright-field (BF) scanning transmission electron microscopy (STEM) image in Fig. 1c shows a nearly homogeneous distribution of precipitates both in the grain interiors and at grain boundaries, with sizes of several hundred nanometers. The BF STEM analysis (Fig. 1c) also shows the formation of the second type of precipitates (identified as κ-carbides) together with the B2 phase, either by covering the interfacial region between B2 phase and γ matrix (white arrows) or with close contact to B2 precipitates of similar size (black arrows).

To further characterize the nanoprecipitates at atomic scales, we performed high-resolution annular bright-field (ABF) STEM and atom probe tomography (APT) analysis. Figure 1d shows a representative B2 particle surrounded by the κ-carbide phase. The magnified images in Fig. 1e and f show the atomic structure of both phases. The ordered structure of B2 is further confirmed by the superlattice reflections along the <001> zone axis in the inset of Fig. 1e, and the ordered structure of κ-carbide is revealed by superlattice reflections along the <110> zone axis from FFT (fast Fourier transform) patterns, as shown in the inset of Fig. 1f. The three-dimensional morphology and chemical compositions of the precipitates are revealed by APT analysis (Fig. 1g). The atomic maps for each element (Fig. 1g) and one-dimensional compositional profiles (Fig. 1h) show that Ni and Al are enriched in the B2 particle and C partitions into the adjacent κ-carbide, confirming the precipitation of B2 and κ-carbide in terms of their respective chemical compositions. Energy dispersive X-ray spectroscopy (EDS) maps in Fig. 1i reveal the co-existence of both (Ni, Al)-rich B2 and C-rich κ-carbide precipitates with similar sizes, confirming the other type of topology observed in this dual-nanoprecipitation system.

Based on the compositional analysis obtained from APT, we estimated the SFE of the austenitic γ matrix by thermodynamic calculations[21–23], suggesting a SFE value of ~79 mJ/m² (see "Methods") for the γ matrix of our CCS, which falls into the SFE range for typical austenite-based Fe–Mn–Al–C lightweight steels[13]. Such a SFE range is usually far too high to activate mechanical twinning in austenitic materials under tensile loading, as shown by multiple studies[13,18].

### Deformation twining under bulk quasi-static tension
We conducted tensile tests on the material using bulk samples at a relatively low strain rate of $1 \times 10^{-3}\,\mathrm{s}^{-1}$ at room temperature (see "Methods"). A typical true stress–strain curve is shown in Fig. 2a (corresponding engineering stress–strain curve is shown in Supplementary Fig. 1), which exhibits an ultrahigh true tensile strength close to 1.9 GPa. Such high strength of our CCS significantly outperforms those of previously designed lightweight steels with similar SFEs (see Fig. 2a). Characterization of the deformation substructures reveals planar slip of the dislocations in the austenitic matrix at a low local strain of ~2% (see Supplementary Fig. 2), which originates from the high lattice friction stress and the planar dislocation cores[14,24]. As the plastic strain increases to ~11%, a remarkably higher density of dislocations is seen, which cut through the κ-carbides and are subsequently pinned by B2 particles (see Supplementary Fig. 2). In addition to the combination of the cutting and bypassing mechanisms,

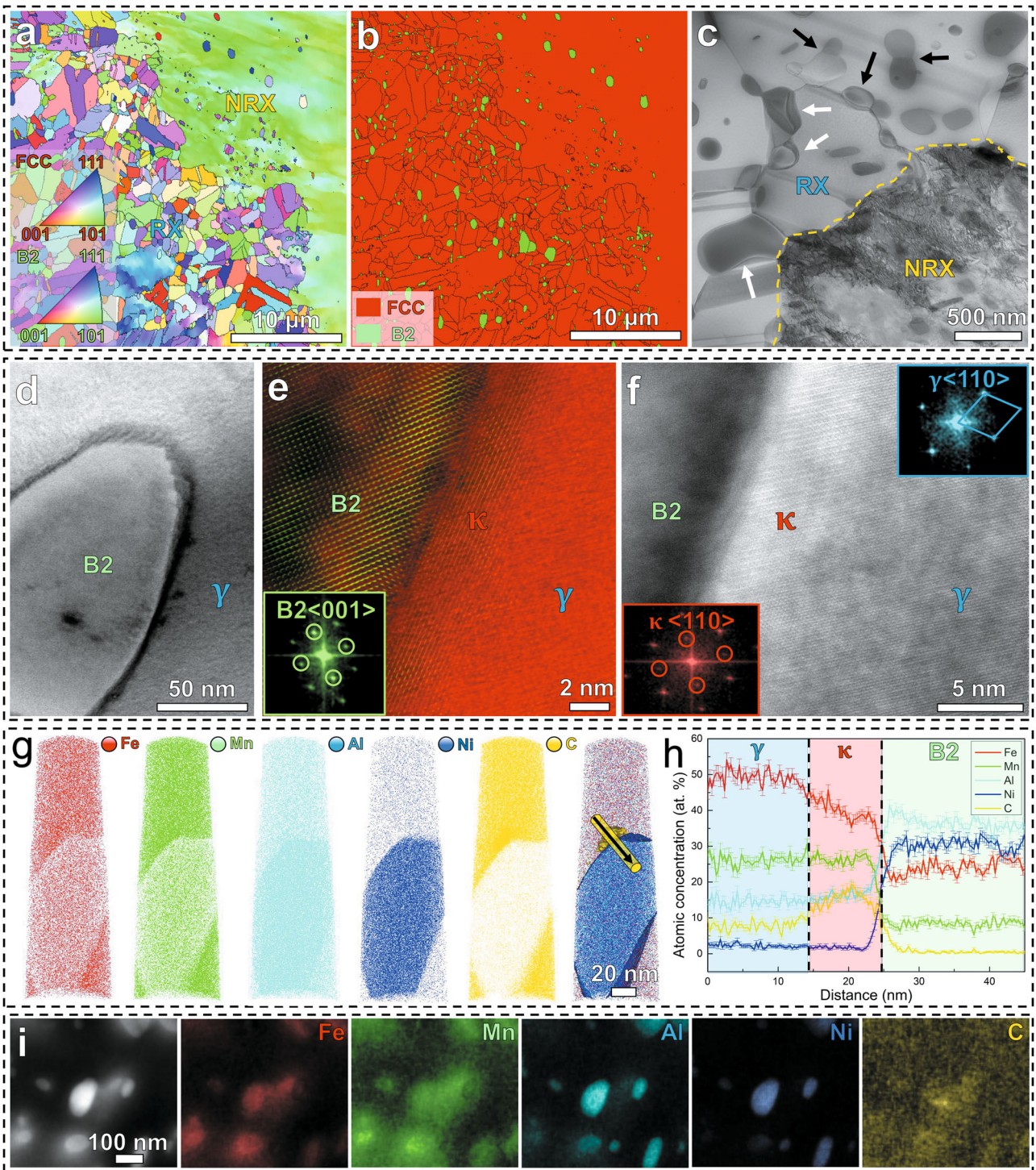

**Fig. 1 Microstructures of the CCS. a** EBSD IPF map. **b** EBSD phase map. **c** BF STEM image. The IPF map shows the existence of both, recrystallized (RX) and non-recrystallized (NRX) regions. The phase map indicates the formation of B2 precipitates in the austenite matrix, while the BF STEM image shows the formation of the other precipitate (κ-carbide) together with B2 phase. Black lines in **b** indicate high angle grain boundaries. Black arrows and white arrows in **c** indicate two types of dual-nanoprecipitation. **d** ABF image and **e**, **f** corresponding Zoom-in images. The inserted FFT patterns confirm structures of B2, κ-carbide and γ matrix. **g** Atomic maps for individual elements and the one highlighted by iso-composition surfaces of 20 at. % Ni and 12 at. % C. **h** One-dimensional compositional profiles along the black arrow in **g** showing the chemical compositions of each phase. **i** HAADF STEM image and EDS maps showing the close contact of κ-carbide and B2 precipitates with similar sizes. IPF, BF, ABF, FFT, HAADF, and EDS refer to "inverse pole figure", "bright-field", "annular bright-field", "fast Fourier transform", "high-angle annular dark-field", and "energy dispersive X-ray spectroscopy", respectively.

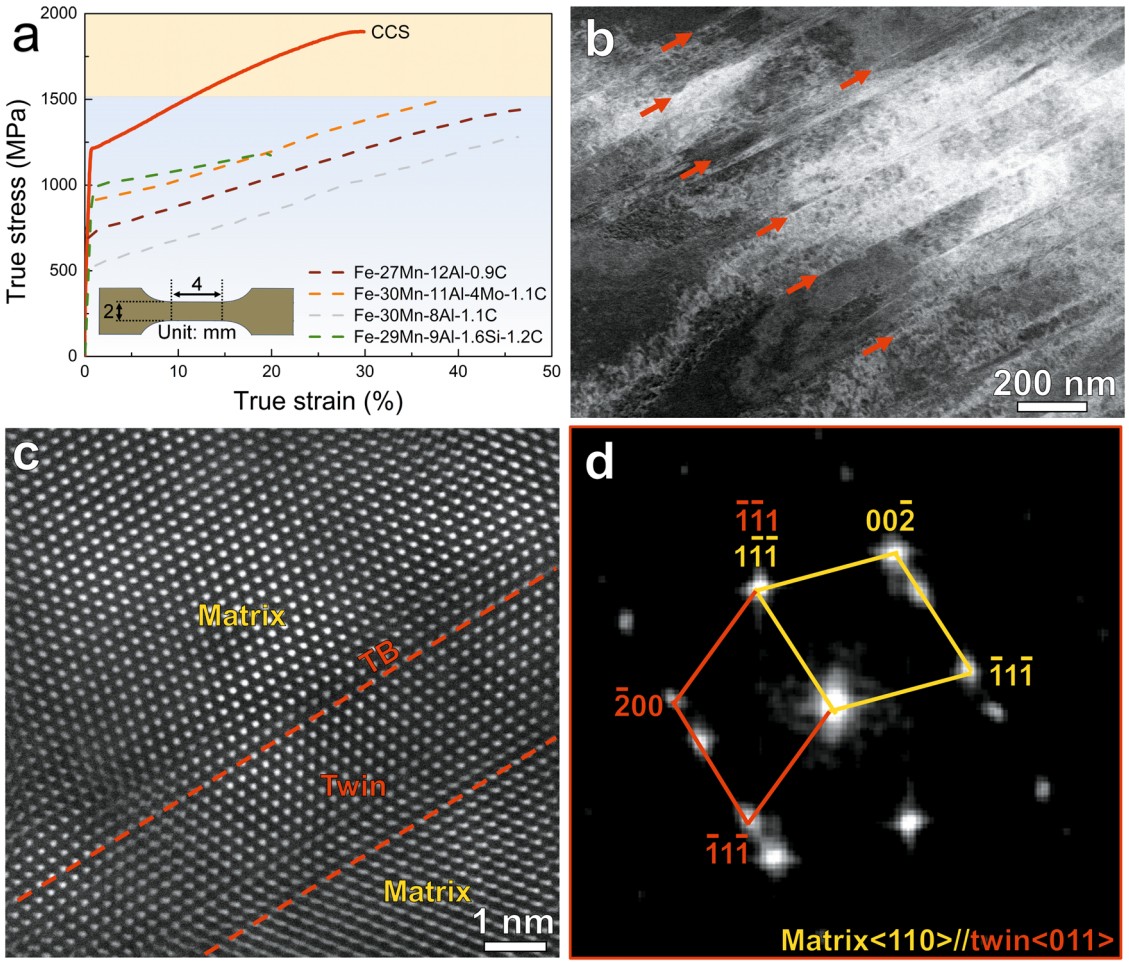

**Fig. 2 Mechanical performance of the CCS and deformation microstructure. a** True tensile stress–strain curve for the CCS under tension at a strain rate of $10^{-3}$ s$^{-1}$. The inset schematically shows the geometry of the bulk tensile sample. The true stress–strain curves of four typical lightweight steels with similar SFEs to our CCS (Fe-27Mn-12Al-0.9C[40], Fe-30Mn-11Al-4Mo-1.1C[41], Fe-30Mn-8Al-1.1C[42], and Fe-29Mn-9Al-1.6Si-1.2C[16], in wt. %) are shown for comparison. **b** Deformation substructures of the CCS at a local strain of 70%. Red arrows indicate the nano-twins. **c** High-resolution HAADF image showing the atomic structure of deformation nano-twinning. TB indicates the twin boundary. **d** FFT pattern confirming the twinning structure.

dislocations are also observed inside the B2 particles, suggesting the co-deformation of B2 and matrix at medium strain levels. Overall, the plastic deformation proceeds via dislocations at low and medium strains (see Supplementary Fig. 2), similar as in other lightweight steels[13].

At higher strain levels, we detect mechanical twinning in our alloy. We lifted out a specimen in a region with a local strain of ~70% (near the fracture surface; see Supplementary Fig. 3) for STEM analysis. In addition to a high density of dislocations, the high-angle annular dark-field (HAADF) STEM image in Fig. 2b shows that multiple parallel thin plates have formed at this strain stage. The high-resolution HAADF STEM observation clearly reveals the twinning of atomic planes (Fig. 2c), and the corresponding FFT pattern further shows a <110>matrix//<011>twin system which is the common twin system in FCC alloys (Fig. 2d).

**Formation of twinning via in situ TEM analysis.** To better understand the formation mechanism of these mechanical twins, we performed in situ TEM tensile tests (Fig. 3a–h and Supplementary Video). The starting microstructure of the CCS specimen is shown by both BF (Fig. 3a) and low angle annular dark field (LAADF) (Fig. 3e) images. Apart from the precipitates, both, the recrystallized region with fine grains and the non-recrystallized

region with dislocations are revealed, consistent with the microstructures mapped for the bulk material (Fig. 1c). During the in situ tensile test, dislocation formation and motion dominate the plastic deformation from the early to medium strain stages. LAADF STEM images in Fig. 3b and BF TEM images in Fig. 3f clearly display the dislocation motion at the early stage of deformation, and the density of the dislocations increases with increasing strain.

At the later stage of the in situ deformation experiment in the TEM, a plate with relative dark contrast appears in a region where a crack begins to occur (Fig. 3g), identified as the twinning event by selected area electron diffraction (SAED) analysis (inset in Fig. 3g). It is worth noting that such twinning simultaneously occurs also in other grains that are far away from the crack (Fig. 3 and Supplementary Fig. 5). The deformation twins observed during the in situ TEM tensile test are several hundred nanometers thick, i.e., significantly larger than the twins with thicknesses of only several nanometers, which were found in the bulk sample (Fig. 2). This is because the flow stress is substantially higher in the in situ specimen than that in the bulk tensile testing due to the thin film effect[25,26]. Another interesting feature is that the crack did not initiate at the incoherent interfaces between matrix and B2 particles, and the following growth of the crack went through different grains (Fig. 3d, h), showing an intragranular fracture mode, as further confirmed by EDS maps

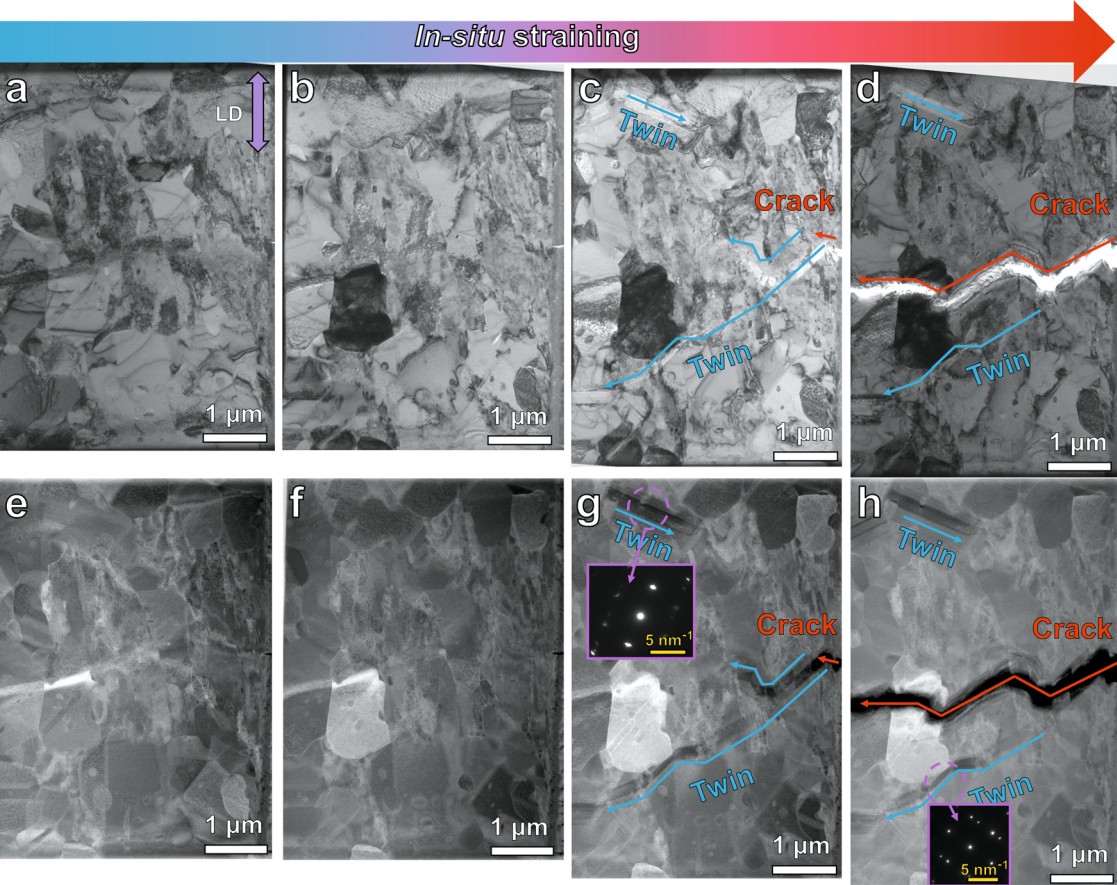

**Fig. 3 In situ BF TEM (a–d) and corresponding LAADF STEM (e–h) observations under tensile test for the CCS. a, e** Starting microstructure of the CCS prior to tensile loading. **b, f** Dislocation slip dominates the early stage of deformation. **c, g** Formation of mechanical twins at various regions when a crack nucleates. **d, h** Cracks propagate into different grains, while fracture at the interfaces among B2 and the austenite matrix is suppressed. The inserted SAED patterns in **g** and **h** reveal twinning structures, and the correspondingly indexed pattern is shown in Supplementary Fig. 4. LD indicates the loading direction.

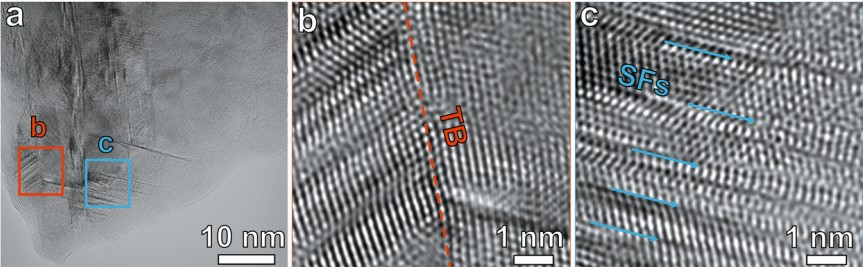

**Fig. 4 TEM images showing the deformation microstructures of the CCS after the in situ tensile test. a** Twins and stacking faults near the cracked regions. **b** Zoom-in image showing the twinning. **c** Zoom-in image showing the stacking faults. "TB" and "SFs" indicate the twin boundary and stacking faults, respectively.

shown in Supplementary Fig. 6. This feature is distinctly different from the crack growth behavior in conventional alloys where tearing and delamination of interfaces between FCC and B2 is observed[27].

We further investigated the sample regions near the crack after the in situ TEM tensile test by subsequent high-resolution TEM analysis. Multiple twins are seen in Fig. 4a, and the high-resolution TEM images reveal twinning structures (Fig. 4b). Multiple stacking faults are also detected after the in situ tensile tests (Fig. 4a). A faulted stacking sequence of atoms in the FCC matrix is revealed by the TEM images in Fig. 4c.

The nucleation of 1/6 <112> Shockley partial dislocations is essential for the formation of deformation twinning (Supplementary Fig. 7). For materials with very high SFEs, dislocation motion proceeds via perfect dislocations 1/2 <110>, since partial dislocations come at the costs of stacking faults and thus require high stresses to form. We estimate that the critical twinning stress in our steel is 1.5–1.7 GPa (see "Methods"). This stress value is much higher than the tensile flow stresses of previously studied lightweight steels with similar SFEs, yet with maximum tensile stress levels below 1.5 GPa (Fig. 2a). The ultrahigh true tensile stress of our steel (up to 1.9 GPa, see

Fig. 2a) reaches the required high critical twinning stresses, thus leading to deformation twinning in this material, irrespective of its high SFE.

## Discussion

The resulting excellent mechanical properties of the present CCS originate from the specific alloy design strategy that is guided by two main ideas. (1) We adopt the HEA concept to redesign lightweight steels by enhancing their compositional complexity, achieving the previously unattainable dual-nanoprecipitation system in our CCS. These nanoprecipitates substantially strengthen the alloy through dislocation bypassing and cutting[20], leading to high strain hardening[28,29]. (2) At higher deformations, the high stress level, which arises from the increasing dislocation densities and the associated complex interactions between dislocations and nanoprecipitates (Supplementary Fig. 2), allows activating the previously inaccessible deformation-induced nanotwinning mechanism in the austenite matrix. The formation of nanotwins in turn enables further strain hardening and toughening reserves during the later stages of deformation, which counteract softening and strain localization, as seen in some HEAs[19,30,31] and lightweight steels[32,33]. The fact that this alloy design strategy is successful, conveys an important and general lesson: materials with high SFEs can be re-modelled for the activation of mechanical twinning and its exploitation for strengthening, provided the required high strength levels can be achieved by other strain hardening mechanisms. Our work thus revises the common understanding that mechanical twinning is inaccessible in high SFE materials under quasi-static loading conditions, by showing that mechanical twinning was triggered as a regular bulk deformation mechanism when the activation stresses for competing deformation mechanisms (usually dislocation motion) are high enough without reaching the fracture strength.

In the current CCS the enhanced strength and strain hardening of the austenite matrix, resulting from deformation twinning, together with the co-deformation of the B2 precipitates (Supplementary Fig. 2), helps to avoid stress localization at the incoherent interfaces among austenite/κ-carbides and B2 phases. In addition, the formation of nanotwins can impede the propagation of cracks, thus contributing a significant toughening effect[31,34]. All of these mechanisms delay the nucleation and propagation of cracks which are commonly seen in B2 strengthened FCC-based alloys and significantly detrimental to an alloy's ductility[27]. Therefore, excellent tensile elongation (~30%) is achieved in our CCS, even at such a high stress level (~1.9 GPa), where alloys are otherwise more brittle.

In summary, we demonstrate the activation of deformation nanotwinning during quasi-static tensile testing of a bulk FCC-based lightweight CCS (at the mm scale) with very high SFE (~79 mJ/m$^2$), an interesting mechanism combination of fundamental relevance in the field of plastic deformation of metallic materials. Owing to the dual-nanoprecipitation of κ-carbide and B2 phases, enabled by our specific alloy design strategy, the CCS shows very high tensile stress, thus reaching the critical twinning stress, which had been so far inaccessible for conventional lightweight steels. The high stress twinning phenomenon provides remarkable strain hardening during the later stages of deformation, which suppresses the nucleation of cracks at the incoherent interfaces between matrix and B2 phases and thus, leads to excellent ductility of the material. Our study thus shows a promising design strategy, by triggering previously unattainable deformation mechanisms in high-performance structural materials with high SFE, for enhancing their mechanical properties.

## Methods

**Materials.** The actual chemical composition of the CCS is Fe-26.4Mn-15.9Al-4.5Ni-4.9C in at. %, corresponding to Fe-29.6Mn-8.8Al-5.2Ni-1.2C in wt. %, obtained from wet-chemical analysis. An induction furnace was used to cast the CCS under Ar atmosphere. The homogenization was performed at 1200 °C for 1 h protected by Ar, followed by the water quench. Subsequently, hot-rolling at 1100 °C and cold-rolling at room temperature were applied to reduce the thickness from 10 to 1 mm. Final annealing was conducted at 900 °C for 3 min under vacuum using a DIL805A/D dilatometer, at a heating rate of 50 °C/s and a cooling rate of 220 °C/s.

**Characterization.** Samples for scanning electron microscopy (SEM) measurements were mechanically polished, with subsequent fine polishing using silica oxide suspension. EBSD mapping was conducted on a Zeiss Sigma 300 SEM instrument.

Twin-jet electropolishing and focused ion beam (FIB) were employed to prepare the TEM samples. A Struers TenuPol-5 was applied to polish specimens using 30% nitric acid in methanol using at a voltage of ~10 V and a temperature of −30 °C. The SAED, transmission electron microscopy (TEM), scanning transmission electron microscopy (STEM), and EDS analysis were carried out by using both image aberration-corrected and probe aberration-corrected TEM (FEI Titan Themis) at 300 kV. For HAADF STEM imaging, a probe semi-convergence angle of 17 mrad and inner and outer semi-collection angles ranging from 73 to 200 mrad were used. For LAADF imaging, a probe semi-convergence angle of 17 mrad and inner and outer semi-collection angles from 14 to 63 mrad were used.

The in situ deformation was realized using a custom-designed Cu tensile dog-bone holder, which facilitates the simultaneous acquisition of the deformation and real-time TEM/STEM images. With the TEM tensile dog-bone holder, containing the TEM lamella, loaded into a Gatan holder, the in situ tensile deformation was performed at room temperature and at constant strain rate.

Needle-shaped samples for APT were produced by a dual FIB-SEM system (FEI Helios NanoLab 600i). APT measurements were performed on a CAMECA LEAP 5000 XR local electrode instrument in voltage-pulsing mode using a repetition rate of 200 kHz, a pulse fraction of 15%, and a specimen temperature of −203 °C. The APT data were analyzed using the commercial IVAS 3.8.4 software. It should be noted that Fe$_{54}^{2+}$ and Al$_{27}^{1+}$ overlap at 27 Da. Thus, we did the decomposition analysis[35], which distributed 35 % of the composition to Al and 65% of the composition to Fe.

Tensile tests were conducted at room temperature at an initial strain rate of $10^{-3}$ s$^{-1}$ by a Kammrath & Weiss tensile stage, using dog-bone shaped specimens with a gauge length of 4 mm, a width of 2 mm, and a thickness of 1 mm. We track the local strain evolution during tensile testing based on a digital image correlation (DIC) method by using an Aramis system (GOM GmbH).

**Calculation of the stacking fault energy.** The SFE ($\Gamma$) of the austenite matrix in our CCS was evaluated based on the well-established thermodynamic approach[21,22]:

$$\Gamma = 2\rho\Delta G^{\gamma\rightarrow\varepsilon} + 2\sigma \tag{1}$$

where $\rho$ is the molar surface density along {111} planes, $\Delta G^{\gamma\rightarrow\varepsilon}$ is the molar Gibbs energy of the phase transformation from austenite to martensite, and $\sigma$ is the interfacial energy per unit area of the phase boundary. The change of the molar Gibbs energy during phase transformation can be expressed as[21,36]:

$$\begin{aligned}\Delta G^{\gamma\rightarrow\varepsilon} = &\chi_{FE}\Delta G_{FE}^{\gamma\rightarrow\varepsilon} + \chi_{Mn}\Delta G_{Mn}^{\gamma\rightarrow\varepsilon} + \chi_{Al}\Delta G_{Al}^{\gamma\rightarrow\varepsilon} + \chi_{Ni}\Delta G_{Ni}^{\gamma\rightarrow\varepsilon} + \chi_{C}\Delta G_{C}^{\gamma\rightarrow\varepsilon} \\ &+ \chi_{FE}\chi_{Mn}\Omega_{FEMn}^{\gamma\rightarrow\varepsilon} + \chi_{Fe}\chi_{Al}\Omega_{FEAl}^{\gamma\rightarrow\varepsilon} + \chi_{Fe}\chi_{Ni}\Omega_{FENi}^{\gamma\rightarrow\varepsilon} + \chi_{Fe}\chi_{C}\Omega_{FEC}^{\gamma\rightarrow\varepsilon} \\ &+ \chi_{Mn}\chi_{C}\Omega_{MnC}^{\gamma\rightarrow\varepsilon} + \Delta G_{mg}^{\gamma\rightarrow\varepsilon}\end{aligned} \tag{2}$$

where $\chi$ is the atomic molar fraction and $\Omega$ is the excessive mixing energy. In Fe–Mn–Al–C lightweight steel, the possible interactions among rest elements, such as Mn–Al and Al–C, are assumed to be negligible[36]. $\Delta G_{mg}^{\gamma\rightarrow\varepsilon}$ is the Gibbs energy for the magnetic state of the phase, which is given by

$$\Delta G_{mg}^{\gamma\rightarrow\varepsilon} = G_{mg}^{\varepsilon} - G_{mg}^{\gamma} \tag{3}$$

The Gibbs energy from the magnetic contribution for each phase ($\gamma$ or $\varepsilon$) can be calculated by the equation[21]

$$G_{mg}^{(\gamma\,or\,\varepsilon)} = RTln(\beta^{(\gamma\,or\,\varepsilon)}+1)f(\tau^{(\gamma\,or\,\varepsilon)}) \tag{4}$$

where $R$ is the gas constant, $T$ is the temperature ($T$ = 298 K), $\beta$ is the magnetic moment of phase divided by the Bohr magneton, and $f(\tau)$ is a polynomial function of the scaled Nèel temperature ($\tau = T/T_{Nèel}$)[21]

$$f(\tau) = -\left[\frac{\tau^{-5}}{10} + \frac{\tau^{-15}}{315} + \frac{\tau^{-25}}{1500}\right]/D, \text{ if } \tau>1; \tag{5}$$

$$f(\tau) = 1 - \left[\frac{79\tau^{-1}}{140p} + \frac{474}{497}\left(\frac{1}{p}-1\right)\left(\frac{\tau^3}{6} + \frac{\tau^9}{135} + \frac{\tau^{15}}{600}\right)\right]/D, \text{ if } \tau \leq 1; \tag{6}$$

where $p$ = 0.28 and $D$ = 2.34 for FCC[23]. Thermodynamic parameters for Fe, Mn, Al, and C were taken from ref. [23] for the Fe–Mn–Al–C alloy system, and those for Ni were obtained from the Fe–Cr–Ni alloy system[21]. It should be noted that the

value of the interfacial energy is not sensitive to compositional variations of Fe–Mn–Al–C steels[36]. The possible influence of chemical ordering can be neglected here, since APT analysis (Supplementary Fig. 8) and FFT pattern analysis (inset in Fig. 1f) showed no sign of ordering in the FCC matrix.

By using all values and functions listed in Supplementary Table 1, the SFE is calculated as 79 mJ/m². This calculated high value of the SFE is also supported by the zoom-in LAADF-STEM observations which were conducted at a strain of 2%, as shown in Supplementary Fig. 2, where no dissociation of dislocations was detected. Moreover, the thermodynamic model used here has been validated by SFEs measured by various other methods, as reported before in the literature, including X-ray diffraction (XRD), TEM and neutron diffraction, for several Fe–Mn–Al–C steels (see Supplementary Table 2). As summarized in a review about TWIP steels[18], deformation twins form at the SFE range of ~10–50 mJ/m². By comparison, the SFE of our steel is with 79 mJ/m² far higher than that of representative TWIP steels.

**Calculation of the critical stress for twinning**. The critical stress for twinning ($\sigma_t$) can be estimated by a mean-field model[6,37,38]

$$\sigma_t = M\left(\frac{\Gamma}{b} + \frac{Gb}{D}\right) \quad (7)$$

where $M$ is the Taylor factor for a randomly textured polycrystal ($M = 3.06$ based on the Taylor model and $M = 2.7$ based on the crystal plasticity finite element model[39], $\Gamma$ is the SFE (79 mJ/m²), $G$ is the polycrystal shear modulus (68 GPa ref. [20]), b is the magnitude of the Burgers vector of the partial dislocation (0.147 nm ref. [18]), and $D$ is the grain size. The value of grain size ($D$) is determined to be ~5 μm, by averaging sizes between the recrystallized (RX) and non-recrystallized (NRX) regions. After the calculation, the critical twinning stress for our CCS is around 1.5–1.7 GPa. The high strain hardening from interactions among the dislocations and the dual-nanoprecipitation system plays a significant role in reaching the critical twinning stress in our CCS.

The actual grain size may change as the twinning occurs at a relatively large strain. Since the SFE plays a dominant role in the twinning stress, particularly for the current steel with very high SFE (79 mJ/m²) and high twinning stress (1.5–1.7 GPa), the possible influence from the variation of grain size during deformation is negligible. The dislocation density reached ~3.12 × 10¹⁴ m⁻² at an intermediate bulk sample strain of 11%, at which the originally RX and NRX grains can be hardly distinguished anymore. Given that the twinning occurs at very large strain, the NRX and RX grains show little differences in their respective twinning capability.

## Data availability

All data needed to evaluate the conclusions in the paper are present in the paper and/or the Supplementary Materials. Additional data related to this paper may be requested from the corresponding authors.

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

## Acknowledgements

Z.W. would like to acknowledge the support from State Key Laboratory of Powder Metallurgy, Central South University, Changsha, China, and Changsha Municipal Natural Science Foundation (kq2202091). Z.L. would like to acknowledge the support from the National Natural Science Foundation of China (Grant No. 51971248) and the Natural

Science Foundation of Hunan province in China (Grant No. 2021JJ10056). Financial support of the German Research Foundation within the Priority Programme 2006 (Compositionally Complex Alloys – High Entropy Alloys) is acknowledged. M.S. would like to acknowledge the support from the National Natural Science Foundation of China (Grant No. 51971247). W.L. is grateful for the financial support from the Shenzhen Science and Technology Program (JCYJ20210324104404012) and the open research fund of Songshan Lake Materials Laboratory (2021SLABFK05).

## Author contributions

Z.W., W.L., and Z.L. conceived the project. Z.W. prepared the materials, performed the tensile tests, and conducted the SEM/EBSD and APT characterization. W.L. and F.A. conducted the TEM/STEM and in situ characterization. M.S., D.P., and D.R. contributed to the data analysis. Z.W., W.L., D.R., and Z.L. wrote the manuscript. All authors contributed to the discussion of the results and commented on the manuscript.

## Competing interests

The authors declare no competing interests.
