## [Peer Review File · Nature Communications]

Title: High stress twinning in a compositionally complex steel of very high stacking fault energyReviewers' comments:

Reviewer #1 (Remarks to the Author):

The article by Wang et al. introduces a novel approach to improving strength and ductility of steels by means of deformation twinning even for steels with compositions characterized by a high stacking fault energy (SFE). At first glance, this strategy appears counter-intuitive, but it turns out to be very successful, at least in the particular case of the Fe-26Mn-16Al-5Ni-5C steel with the SFE of about 79 mJ/m² investigated in the article. This magnitude of the SFE is well above the SFE range in which a face-centered cubic steel would exhibit deformation twinning. The authors show, however, that due to deformation-induced precipitation in this steel with a complex composition the strain hardening drives the flow stress to such levels that the critical stress for the onset of twinning is exceeded.

Through a rigorous analysis of the deformed microstructures based on the state-of-the-art electron microscopy (both TEM and STEM), EDS, and atom probe analysis, the authors identified the precipitated particles as the B2 phase and the kappa-carbides and showed that this dual nanoprecipitation can be gainfully employed as a way to achieve an exceptional combination of mechanical properties. The excellent tensile elongation of nearly 30% paired with a very high strength level of about 1.9 GPa is a remarkable outcome of this study. What is more, the authors suggest that this strategy opens up a new paradigm in alloy design with implications far beyond the singular result they obtained.

The article is well written and conveys the main message of the authors very clearly. The quality of the experimental data is high and leaves no doubt that the results are genuine and compelling.

This article will be a good read for a broad readership of Nature Materials and its acceptance for publication is recommended.

Reviewer #2 (Remarks to the Author):

The current work shows an interesting example that TWIP effect occurs in high-SFE steel. Besides, such phenomena makes the true stress reach 2000 MPa for the steel. The concerns for publication of this work are listed below.

1. Novelty in materials property has been published.

The authors has published this materials in the other letter-type journal (Wang et al., Sci. Adv. 2020; 6: eaba9543). In that work, they claimed the novel property comes from dual precipitates but now comes from TWIP. Such publication history makes uncertainty in the research quality.

2. Science in materials behavior seems not new.

TWIP effect in high-SFE steel has been published in (Z.H. Lai et al., Acta Mat., 210, 116814, 2021) and (I. Gutierrez-Urrutia et al., Acta Mat., 60, 5791–5802, 2012). In Lai's work, the SFE was about 70 mJ/m²

calculated by Curtze's method and the principle was clearly explained.

These above are two primary concerns about this manuscript.

3. Twinning stress

The equation used to estimate twinning stress did not consider the work hardening behavior. Plastic deformation could make the proposed confinement length, D , shorter (see Lai's work). Besides, how did the author estimate the value D to be 5 micrometers? Did the author try other method to estimate SFE?

4. Crack propagation

The materials show very limited ductility in post elongation. Is crack propagation path really important here?

5. Dislocation density

The materials yield at 1200 MPa (or above) but strain hardened to 2000 MPa. What is the change in dislocation density? How is the ability of this materials to maintain a high dislocation density?

6. Materials homogeneity

There are RX and NRX grains in the materials. Do they have different twinnability? How are the change in dislocation density for these two kind of grains in deformation?

7. Color key is required for inverse pole figure in Fig. 1.

8. Although many advanced characterizations were applied here, the authors should still show the formation of deformation twins. Is there any evidence of SF emitted from substructures in the steel?

9. Why did lower twin in Fig. 3 propagate in similar direction across several grains? What are the misorientations for these grains?

10. Did the author find short-range ordering in gamma matrix?

Reviewer #3 (Remarks to the Author):

The formation of deformation twins for an alloy with the stacking fault energy (SFE) of $\sim 79 \text{ mJ.m}^{-2}$ is the central theme of the manuscript. If true, the result carries a very high significance because it challenges the status quo where the deformation twinning is possible only for SFE below $\sim 50 \text{ mJ.m}^{-2}$. For this claim, the authors have entirely relied on thermodynamic calculation for the estimation of SFE of the compositionally complex steel studied in this work. There is no discussion on the accuracy of SFE estimation following the procedure outlined. There is also no comment on the choice of thermodynamic

parameters which have been taken from literature on other steels with different compositions and utilized for the present system. In literature, one can find examples of different values of SFE being reported for steels with very similar compositions using thermodynamic calculation. For example, Shun et al. [1] reported an SFE of 50.1 mJ/m^2 for Fe-30.9Mn-2.7Al-0.96C (wt.%) steel whereas Gutiérrez-Urrutia and Raabe [2] reported a value of 63 mJ/m^2 for Fe-30.5Mn-2.1Al-1.2C (wt.%) steel (Note: it is interesting to note that in ref [2] included below, Gutiérrez-Urrutia and Raabe have shown the presence of deformation twinning for Fe-30.5Mn-2.1Al-1.2C (wt.%) steel with the SFE of 63 mJ/m^2 which is greater than 50 mJ/m^2). Similarly, the choice of interfacial free energy has been taken as 9 mJ/m^2 . Others in literature as well have used the same value for interfacial free energy for the steels with different chemical compositions (for example, Ref. [28] in the manuscript). Does it mean chemical compositions will not have any impact on the interfacial free energy?

Because of the significance of the result, the authors should have also estimated the SFE of the steel with alternative means, that is, X-ray diffraction and transmission electron microscopy to check what values of SFE the experimental techniques provided and how it compared against the thermodynamically estimated SFE.

[1] Shun, T., Wan, C. M., & Byrne, J. G. (1992). A study of work hardening in austenitic Fe-Mn-C and Fe-Mn-Al-C alloys. *Acta metallurgica et materialia*, 40(12), 3407-3412.

[2] Gutiérrez-Urrutia, I., & Raabe, D. (2012). Multistage strain hardening through dislocation substructure and twinning in a high strength and ductile weight-reduced Fe-Mn-Al-C steel. *Acta Materialia*, 60(16), 5791-5802.

Response to Reviewers' Comments

NCOMMS-21-36513-T

We would like to begin by thanking the editor and reviewers for the valuable suggestions and comments. Our response is structured as follows: The comments from the reviewers are copied below (black, italic font). For each comment, we present a response and the corresponding manuscript modifications (blue font). The amended manuscript is enclosed. The changes therein are shown in red font.

Reviewer #1

The article by Wang et al. introduces a novel approach to improving strength and ductility of steels by means of deformation twinning even for steels with compositions characterized by a high stacking fault energy (SFE). At first glance, this strategy appears counter-intuitive, but it turns out to be very successful, at least in the particular case of the Fe-26Mn-16Al-5Ni-5C steel with the SFE of about 79 mJ/m² investigated in the article. This magnitude of the SFE is well above the SFE range in which a face-centered cubic steel would exhibit deformation twinning. The authors show, however, that due to deformation-induced precipitation in this steel with a complex composition the strain hardening drives the flow stress to such levels that the critical stress for the onset of twinning is exceeded.

Through a rigorous analysis of the deformed microstructures based on the state-of-the-art electron microscopy (both TEM and STEM), EDS, and atom probe analysis, the authors identified the precipitated particles as the B2 phase and the kappa-carbides and showed that this dual nanoprecipitation can be gainfully employed as a way to achieve an exceptional combination of mechanical properties. The excellent tensile elongation of nearly 30% paired with a very high strength level of about 1.9 GPa is a remarkable outcome of this study. What is more, the authors suggest that this strategy opens up a new paradigm in alloy design with implications far beyond the singular result they obtained.

The article is well written and conveys the main message of the authors very clearly. The quality of the experimental data is high and leaves no doubt that the results are genuine and compelling.

This article will be a good read for a broad readership of Nature Materials and its acceptance for publication is recommended.

Response: We are grateful to the reviewer for the recognition of the novelty of our work and for the clear recommendation for the publication of our work. We have further improved the manuscript based on the editor's and reviewers' comments.

Reviewer #2

The current work shows an interesting example that TWIP effect occurs in high-SFE steel. Besides, such phenomena makes the true stress reach 2000 MPa for the steel. The concerns for publication of this work are listed below.

Response: We thank the reviewer for carefully reading our paper and for making useful comments. We have addressed all comments and revised the manuscript, as shown below in detail.

1. Novelty in materials property has been published.

The authors has published this materials in the other letter-type journal (Wang et al., Sci. Adv. 2020; 6: eaba9543). In that work, they claimed the novel property comes from dual precipitates but now comes from TWIP. Such publication history makes uncertainty in the research quality.

Response: Thanks for the comment. We regret that the original manuscript was not clear enough, which might lead to a misunderstanding about our novelty claims. The dual-nanoprecipitation mechanism reported in our previous paper (Wang et al., Sci. Adv. 2020; 6: eaba9543) does not contradict the here observed TWIP phenomenon in any way. Instead, the strengthening effect from the dual precipitates plays an essential role in activating the twinning mechanism, in that it provides the required shift in the overall material's strength level to make the twinning activation stress accessible. This means that there is absolutely no uncertainty in the research result and its interpretation. The twinning actually needs the high strength level from the precipitates and we have made this more clear in the revision. We specifically explain it in the following.

First, the model alloys studied in our previous work and in the current work are different. The model alloy in our previous paper (Wang et al., Sci. Adv. 2020; 6: eaba9543) was annealed at 800 °C, while in the current manuscript, the alloy was annealed at 900 °C. The different processing procedures lead to distinctly different microstructures, e.g., fraction of recrystallized regions (8 % vs. 57 %), grain size in recrystallized regions (0.35 μm vs. 1.5 μm), and size of dual-precipitates (~4-6 nm vs. ~100-300 nm).

More importantly, the focus of our previous paper (Wang et al., Sci. Adv. 2020; 6: eaba9543) is to introduce dual-nanoprecipitation into the design strategy of novel lightweight compositionally complex steels (or high-entropy steels). The deformation microstructures are shown to prove the

combined effects of particle dislocation shearing and particle bypassing of dislocations, due to the dual-nanoprecipitation. In the current study, we apply these previous findings to achieve the otherwise unattainable TWIP effect in the steel with very high SFE. The reason is that the ultrahigh stress arising from the dual-nanoprecipitation shifts the flow strength of the material into a regime where the mechanical twinning can be activated. The state-of-the-art characterization, including the high-resolution *in-situ* TEM/STEM analysis, indicates that enhanced strength and strain hardening of the austenite matrix resulting from deformation twinning is the key to preventing crack nucleation and crack propagation at the interfaces between austenite matrix and precipitates. We have added new experimental data and calculation results in the revised manuscript to confirm this point, as described in detail in the following response. With the aid of the TWIP effect in the present study, we are able to make full use of the strengthening effect from precipitates, while at the same time diminishing the adverse effect on ductility as much as possible. In short, our previous work offers a foundation for our current study to explore and realize the novel TWIP effect in the steels with very high SFE.

In the present study, the TWIP effect achieved in our steel with very high SFE produces a profound and lasting impact on the design of structural materials. For materials with very high SFE, such as Al and Ni, the available strain-hardening mechanisms remain confined to dislocations and their interactions with grain boundaries and precipitates. The important TWIP effect can be usually not reached, due to the lack of sufficiently high overall flow stress levels. Once proof is given that deformation twinning can occur in such materials (even if it occurs under extreme conditions), very high attention has been attracted in these high SFE materials. In these cases (Fig. R1b and R1c below), deformation twinning was achieved in very fine sized specimens (usually at the μm scale) under compression (and sometimes under very high strain rates). However, we reported that deformation twinning occurs in bulk materials (at the mm scale) with high SFE under quasi-static tensile loading (Fig. R1a below). The specimen size used in our work is straightforward to scale up since materials were produced in line with widely applied metallurgical process routines used in industry. The activation of twinning under quasi-static tension in our work suggests that the TWIP effect can be employed to increase both, strength and ductility for engineering applications of materials with high SFE. Our work presents a significant step forward in applying twinning to design high-performance materials for industries.

Fig. R1 Comparison of pathways to achieve deformation twinning in materials with very high SFE. (a) Our steel probed in the current manuscript under quasi-static tensile conditions. (b) Al thin film under high velocity projectile impact (S. Xue *et al.*, Nature Commun., 2017, 8: 1653). (c) Ni micro-pillar under compression (F. Duan *et al.*, Sci. Adv. 2021, 7, eabg5113).

Modification: We revised the title to highlight the uniqueness of the twinning mechanism in our steel with high SFE: “High stress twinning in a compositionally complex steel of very high stacking fault energy”. In addition, we revised the manuscript to clearly point out that the strengthening effect achieved by the dual-nanoprecipitation is the foundation for the occurrence of mechanical twinning: “This surprising finding of high stress twinning (page 2); producing the high

strength required to activate mechanical twins (page 4); which arises from the dual-nanoprecipitation (on page 8); The high stress twinning...(page 9); The high strain hardening from interactions among the dislocations and the dual-nanoprecipitation system plays a significant role in reaching the critical twinning stress in our CCS (on page 21).” On page 8, we noted that the tensile testing has been conducted under quasi-static conditions and the size of the specimen is of mm scale, to better convey the significance of the observed TWIP effect in the current study.

2. Science in materials behavior seems not new.

TWIP effect in high-SFE steel has been published in (Z.H. Lai et al., Acta Mat., 210, 116814, 2021) and (I. Gutierrez-Urrutia et al., Acta Mat., 60, 5791–5802, 2012). In Lai's work, the SFE was about 70 mJ/m² calculated by Curtze's method and the principle was clearly explained.

Response: We appreciate the concerns raised by the reviewer. Although the two papers noted by the reviewer show a TWIP effect during deformation, the corresponding steels presented in these papers still can be categorized as typical TWIP steels. Their SFEs are near to the upper limit for the TWIP effect in typical TWIP steels (around 50 mJ/m²), which thus cannot be regarded as the TWIP effect in high-SFE steels. We explain this issue in detail below.

In Lai's work (Z.H. Lai *et al.*, Acta Mat., 210, 116814, 2021), the SFE of the steel is 49.5 mJ/m², rather than 70 mJ/m². As noted in the abstract of the paper by Lai *et al.*, “The critical design principle lay in the synergetic addition of Al and Si to obtain simultaneously a stacking fault energy of 49.5 mJ/m²”. They attempted to evaluate the SFE based on different approaches, and concluded that the reliable and reasonable result is 49.5 mJ/m². This value was used correspondingly throughout the whole paper (e.g., abstract, Table 5 for discussion, and conclusion). Basically, 3 wt. % of Si was added to replace the Al in Lai's work, which decreased SFE by about 20 mJ/m², leading to a moderate stacking fault energy of 49.5 mJ/m² and triggering the TWIP effect. This TWIP effect in Lai's work is in good agreement with the claim in our manuscript that the SFE regime for twinning is less than ~50 mJ/m². Thus, Lai's paper further supports the novelty of our work.

In Gutierrez-Urrutia's work (I. Gutierrez-Urrutia *et al.*, Acta Mat., 60, 5791–5802, 2012), the SFE of 63 mJ/m² was reported for the steel (Fe–30.5Mn–2.1Al–1.2C, wt. %), however, this SFE value

was overestimated in Gutierrez-Urrutia’s work, as our model gave a lower value of 48 mJ/m². Gutierrez-Urrutia’s paper did not show the calculation procedures and the values of parameters used to estimate the SFE. Hence, we are unable to compare the calculation procedures of SFE in Gutierrez-Urrutia’s work with our work. It is worth noting that our calculations of SFEs are validated by the calculation and experimental results from the literature. As listed in Table R1 below, we compare the calculated SFEs from the literature and by our method for many typical Fe-Mn-Al-C steels, which shows very good consistency. In addition, we added the zoom-in LAADF-STEM observation of dislocations in our compositionally complex steel (Extended Data Fig. 2 in the revised manuscript). No partial dislocation was detected at the initial deformation based on these STEM measurements, which further supports our calculation that the SFE is very high. Moreover, we compared our calculated SFEs of different Fe-Mn-Al-C steels with the values based on experimental measurements, including TEM, XRD and Neutron diffraction, which again showed very good agreement (Table R1 below).

Table R1 Comparison of representative SFEs from literature and values calculated by our model

Compositions (wt. %)	Calculated SFEs from literature (mJ/m ²)		Calculation of SFE by current model (mJ/m ²)
Fe-31Mn-2.7Al-1C	50 [1]		46
Fe-25Mn-4Al-1C	49 [2]		52
Fe-25Mn-2Al-0.6C	35 [2]		30
Fe-22Mn-6Al-0.6C	51 [3]		53
Fe-22Mn-4Al-0.6C	43 [4]		41
Fe-22Mn-3Al-0.6C	39 [4]		35
Fe-22Mn-1Al-0.6C	28 [4]		23
Fe-18Mn-0.6C	14 [5]		14
Fe-18Mn-1.5Al-0.6C	28 [5]		24
Fe-15Mn-0.6C	12 [5]		11
Fe-15Mn-1.5Al-0.6C	26 [5]		22
--	Experimental SFEs from literature		--
	Methods	SFE (mJ/m ²)	
Fe-18Mn-0.6C	TEM	13 ± 3 [6]	14
	XRD	17 ± 3 [7]	
	Neutron diffraction	21 ± 4 [8]	
	XRD	19 ± 3 [9]	
Fe-18Mn-1Al-0.6C	XRD	30 ± 2 [7]	20
Fe-18Mn-1.5Al-0.6C	TEM	30 ± 10 [6]	24
	Neutron diffraction	29 ± 3 [8]	
Fe-18Mn-2Al-0.6C	XRD	36 ± 2 [7]	27
Fe-18Mn-3Al-0.6C	Neutron diffraction	44 ± 5 [8]	33

References

- [1] T. Shun, C.M. Wan, J.G. Byrne, A study of work hardening in austenitic Fe-Mn-C and Fe-Mn-Al-C alloys, *Acta Metal. Mater.*, 40 (1992) 3407-3412.
- [2] O.A. Zambrano, Stacking Fault Energy Maps of Fe–Mn–Al–C–Si Steels: Effect of Temperature, Grain Size, and Variations in Compositions, *J. Eng. Mater. Tech.*, 138 (2016) 041010.
- [3] K.-T. Park, K.G. Jin, S.H. Han, S.W. Hwang, K. Choi, C.S. Lee, Stacking fault energy and plastic deformation of fully austenitic high manganese steels: Effect of Al addition, *Mater. Sci. Eng. A*, 527 (2010) 3651-3661.
- [4] A. Dumay, J.P. Chateau, S. Allain, S. Migot, O. Bouaziz, Influence of addition elements on the stacking-fault energy and mechanical properties of an austenitic Fe–Mn–C steel, *Mater. Sci. Eng. A*, 483-484 (2008) 184-187.
- [5] J.-K. Kim, B.C. De Cooman, Stacking fault energy and deformation mechanisms in Fe-xMn-0.6C-yAl TWIP steel, *Mater. Sci. Eng. A*, 676 (2016) 216-231.
- [6] J. Kim, S.-J. Lee, B.C. De Cooman, Effect of Al on the stacking fault energy of Fe–18Mn–0.6C twinning-induced plasticity, *Scr. Mater.*, 65 (2011) 363-366.
- [7] J.E. Jin, Y.K. Lee, Effects of Al on microstructure and tensile properties of C-bearing high Mn TWIP steel, *Acta Mater.*, 60 (2012) 1680-1688.
- [8] J.S. Jeong, W. Woo, K.H. Oh, S.K. Kwon, Y.M. Koo, In situ neutron diffraction study of the microstructure and tensile deformation behavior in Al-added high manganese austenitic steels, *Acta Mater.*, 60 (2012) 2290-2299.
- [9] K. Jeong, J.-E. Jin, Y.-S. Jung, S. Kang, Y.-K. Lee, The effects of Si on the mechanical twinning and strain hardening of Fe–18Mn–0.6C twinning-induced plasticity steel, *Acta Mater.*, 61 (2013) 3399-3410.

Furthermore, the work by T. Shun *et al.* (*Acta Metal. Mater.*, 40, 3407-3412, 1992) has reported the SFE of 50 mJ/m² in the TWIP steel with a composition of Fe-31Mn-2.7Al-1C (wt. %), which also provides evidence for the overestimation of SFE in Gutierrez-Urrutia's work. The two steels respectively reported in these two works have very similar compositions, both of which contain ~30 wt. % Mn, ~2 wt. % Al, and ~1 wt. % C. The SFE of 50 mJ/m² (our model gives the very close value as well, see Table R1 above) in Shun's work was also recognized and cited in Gutierrez-Urrutia's paper. As noted on page 5798 of Gutierrez-Urrutia's paper, "Using thermodynamical data available for the FeMnAlC system [3,9,14,50,52], we estimate a SFE of 63 mJ/m² for the Fe–30.5Mn–2.1Al–1.2C (wt.%) alloy. This value is close to the reported SFE of 50 mJ/m² in a Fe–31Mn–2.7Al–1.0C (wt.%) alloy [21]." All this evidence indicates that the SFE of Gutierrez-Urrutia's steel is at the upper limit for the TWIP effect in conventional steels (~50 mJ/m²), which is much lower than that of our compositionally complex steel (~79 mJ/m²).

Therefore, the two papers noted by the reviewer do not diminish the novelty of our work, while instead, they further prove the significance of deformation twinning in compositionally complex steels with very high SFE.

We are grateful that the comments from the reviewer reminded us that the deviation of SFE values existed depending on parameters used in calculation models. Thus, in the revised manuscript, we added an approximation character in front of the SFE values to avoid any possible misunderstanding.

Modification: On page 1, the character of “~” was added in front of the SFE values. On page 3, $> 50 \text{ mJ/m}^2$ is replaced by the upper limit for twinning is $\sim 50 \text{ mJ/m}^2$.

These above are two primary concerns about this manuscript.

3. Twinning stress

The equation used to estimate twinning stress did not consider the work hardening behavior. Plastic deformation could make the proposed confinement length, D , shorter (see Lai's work). Besides, how did the author estimate the value D to be 5 micrometers? Did the author try other method to estimate SFE?

Response: We appreciate the reviewer's comments. The value of confinement length (D) is determined by averaging sizes between the recrystallized and non-recrystallized regions before loading. We agree with the reviewer that the confinement length (D) will become shorter after plastic deformation, but the resulting influence on the twinning stress is very limited. Even when the value of D decreases from $5 \mu\text{m}$ to $0.5 \mu\text{m}$ (the value used in Lai's work), the twinning stress merely increases for $\sim 60 \text{ MPa}$, which is very small given the total critical stress value is up to $\sim 1.7 \text{ GPa}$. For typical TWIP steels with SFEs of $20\text{-}40 \text{ mJ/m}^2$, the variation of confinement length may be critical. But in our compositionally complex steel with very high SFE (79 mJ/m^2), this factor is negligible due to the very high critical twinning stress.

We also estimated the SFE of our steel by zoom-in LAADF-STEM observations (Extended Data Fig. 2 in the revised manuscript). No partial dislocation was detected at the initial deformation

based on these STEM measurements, which confirms the very high SFE of our steel. In addition, we validated our SFE calculations by comparing SFE values of different Fe-Mn-Al-C steels based on our calculation model with the values from experimental measurements, including TEM, XRD and Neutron diffraction, which showed very good agreement (Extended data Table S2 in the revised manuscript).

Modification: On page 21, we complied with the reviewer and added a few new sentences to discuss the influence of the confinement length on the twinning stress: “The actual grain size may change as the twinning occurs at a relatively large strain. Since the SFE plays a dominant role in the twinning stress, particularly for the current steel with very high SFE (79 mJ/m^2) and high twinning stress ($\sim 1.7 \text{ GPa}$), the possible influence from the variation of grain size during deformation is negligible.”

4. Crack propagation

The materials show very limited ductility in post elongation. Is crack propagation path really important here?

Response: Thanks for the comment. Regarding the “post elongation”, the reviewer seems to address the post uniform elongation. Actually, our steel showed $\sim 8\%$ strain after reaching the uniform deformation, indicating good plasticity in the post uniform elongation zone, as shown in the engineering stress-strain curve (see Fig. R2 below). The true stress-strain curves shown in Fig. 2a (manuscript) and Fig. R3 (below) aim to outline the true uniform deformation of the materials.

Fig. R2 Engineering stress-strain curve of our steel. Uniform elongation (ϵ_u) and post uniform elongation (ϵ_p) are marked in the figure.

The crack propagation path is a key issue here because the B2 phase is an intermetallic compound, which is very strong but brittle. As shown in Fig. R3 below, our compositionally complex steel with dual-nanoprecipitation of κ -carbide and B2 exhibits the better combination of strength and ductility than steels with only κ -carbide precipitates. Such an unexpected phenomenon suggests that we utilize the strengthening effect from the B2 phase, yet prevent its usually observed embrittlement effect.

Fig. R3 True stress-strain curves for our steel, Fe-29Mn-9Al-1.6Si-1.2C (Z. Wang, *et al.*, *Acta Mater.* 198, 258–270, 2020) and Fe-30Mn-8Al-1.1C (M.J. Yao *et al.*, *Acta Mater.*, 140, 258-273, 2017).

This is because the significant hardening effect from the dual-nanoprecipitation allows activating the nanotwinning mechanism in the austenite matrix. The nanotwinning in turn unleashes an additional strain hardening effect during the later stages of deformation. This prevents stress localization at the incoherent interfaces among austenite/ κ -carbides and B2 phases, which hinders early nucleation of cracks. As a result, the cracks did not initiate at incoherent interfaces between matrix and B2 particles in our steel, whereas interface tearing is seen in conventional B2 strengthened FCC-based alloys (Y. Liao, I. Baker, Mater. Sci. Eng. A, 528, 3998–4008, 2011), where deformation is only carried by dislocation slip, but not by mechanical twinning. Therefore, excellent uniform tensile elongation (30 %) is achieved in our compositionally complex steel even at such a high strength level.

Modification: We added the engineering stress-strain curve in Extended Data Fig. 1 (page 25) to show the post uniform elongation. On page 7, we revised the manuscript to clarify the different crack propagation behavior in our current study compared to that in the previously reported alloys: “This feature is distinctly different from conventional alloys where tearing and delamination of interfaces between FCC and B2 is observed³².”

5. Dislocation density

The materials yield at 1200 MPa (or above) but strain hardened to 2000 MPa. What is the change in dislocation density? How is the ability of this materials to maintain a high dislocation density?

Response: As commented by the reviewer, dislocation motions for carrying the plastic deformation are indeed very important in this study. We further performed detailed LAADF-STEM observations on our steel at a high strain of ~30 %. The evolution of the dislocation substructures clearly shows that the dislocation density increases dramatically with increasing strain (see Fig. R4 below). The quantitative analysis based on these STEM results shows that the average dislocation density in the recrystallized regions increases from $\sim 2.03 \times 10^{13} \text{ m}^{-2}$ at ~2 % strain to $\sim 1.67 \times 10^{15} \text{ m}^{-2}$ at ~30 % strain, suggesting the high capability of the compositionally complex steel for storing dislocations. Such high dislocation densities also contribute to the high flow stresses, and the activation of twinning during the later deformation stages.

Fig. R4 Dislocation substructures of our compositionally complex steel at strains of 2%, 11%, and 30%. With increasing the strain to 11%, the RX and NRX grains can be hardly distinguished, as the average dislocation density reaches a high value of $\sim 3.12 \times 10^{14} \text{ m}^{-2}$.

Modification: We updated the LAADF-STEM image to show the details of dislocation substructure evolution at different strains in the Extended Data Fig. 2 (page 26). On page 21, we added a sentence to further discuss the critical effect of strain hardening from the dislocations: “The high strain hardening from interactions among the dislocations and the dual-nanoprecipitation system plays a significant role in reaching the critical twinning stress in our CCS.”

6. Materials homogeneity

There are RX and NRX grains in the materials. Do they have different twinnability? How are the change in dislocation density for these two kind of grains in deformation?

Response: Thanks for the comment. The RX and NRX grains did not show a notable difference in twinning ability. As shown in the response to comment #5, the dislocation densities reached a similar value of $\sim 3.12 \times 10^{14} \text{ m}^{-2}$ at a medium strain of 11% for both RX and NRX grains (Fig. R5). Since the deformation twinning occurs at a larger strain and the corresponding dislocation densities in both RX and NRX grains are similarly high, the pre-existed dislocations in the NXR

grains show little influence on the twinning ability. To make this clear, in the revised manuscript, we point out that the pre-existed dislocations in the NRX grains had no significant influence on the twinning ability.

Modification: On page 21, we added several sentences to discuss the twinning ability of RX and NRX grains: “The dislocation density reached $\sim 3.12 \times 10^{14} \text{ m}^{-2}$ at an intermediate bulk sample strain of 11%, at which the originally RX and NRX grains can be hardly distinguished anymore. Given that the twinning occurs at very large strain, the NRX and RX grains show little differences in their respective twinning capability.”

7. Color key is required for inverse pole figure in Fig. 1.

Response: Thanks for the note. We added the color key of both FCC phase and B2 phase for the inverse pole figure map in Fig. 1 (See Fig. R5 below).

Fig. R5 Inverse pole figure in Fig. 1

Modification: We updated Fig. 1a on Page 13.

8. Although many advanced characterizations were applied here, the authors should still show the formation of deformation twins. Is there any evidence of SF emitted from substructures in the steel?

Response: Many thanks for this comment. We did further observations using HRTEM and HAADF-STEM imaging around the fracture area of a tensile tested sample to reveal the formation

mechanism of the twins. As shown in Fig. R6 below, the motion of partial dislocations is the key to promoting the formation of stacking faults and twins during plastic deformation. The twin boundary (TB) observed along the $\{111\}$ plane was separated by the partial dislocation to two parallel lines (TB1 and TB2).

Fig. R6 (a) HRTEM reveals the formation of stacking faults (SFs) in the FCC matrix (marked by red arrows). (b) HAADF-STEM imaging shows the formation mechanism of a mechanical twin in our compositionally complex steel, which is associated with partial dislocation motion (leading partial with Burgers vector $\langle 112 \rangle a/6$ indicated by a black arrow).

Modification: We added this figure to reveal the formation process of twinning in Extended Data Fig. 7 (page 31). We also pointed out the documentation of such mechanism on page 7.

9. Why did lower twin in Fig. 3 propagate in similar direction across several grains? What are the misorientations for these grains?

Response: We thank the reviewer for raising this point. We added zoom-in images for Fig. 3, which show that the twins changed the directions across several grains, rather than the same direction (Fig. R7 below). During the *in-situ* tests, the very high stress in the very thin specimen triggers partial dislocations emission, and the glide across neighboring grains becomes possible by changing directions. These boundaries between grains are normal high angle boundaries, of which the misorientation angles are larger than 15° .

Fig. R7 *In-situ* deformation analysis via LAADF-STEM exhibits the twinning formation in the compositionally complex steel. The red dashed lines represent the grain boundaries. The yellow dashed lines outline the deformation twins generated by tensile loading. Twins pass through the grains and change directions depending on the misorientation angle between grains.

Modification: We added the zoom-in images from *in-situ* TEM tests in Extended data Fig. 5 (page 29) to show the changes in directions of twinning propagation across several grains.

10. Did the author find short-range ordering in gamma matrix?

Response: Thanks for the comment. We further conducted atom probe tomography (APT) analysis in the gamma matrix. Based on APT data, we performed a frequency distribution analysis of atoms by using the chi-squared statistical test (see the histogram in Fig. R8 below). The values of the normalized chi-squared parameter μ ($\mu=0$ means ideal randomness and $\mu=1$ means the complete association of atoms) are very close to 0 for all five elements (Fe, Mn, Al, Ni, and C) in the gamma matrix. This indicates the nearly ideal random distribution of the elements. In addition, the FFT pattern (inset in Fig. 1f) confirms the disordered FCC structure of the gamma matrix. In short, we did not find short-range ordering in the gamma matrix.

Fig. R8 A typical APT map and the corresponding frequency distribution analysis for the gamma matrix in our compositionally complex steel.

Modification: We added this APT analysis as the Extended data Fig. 8 (page 32) to show the atomic distribution and specifically the absence of ordering effects in the gamma matrix. Also, we added a new sentence to address this issue on page 20: “The possible influence of chemical ordering can be neglected here, since APT analysis (Extended Data Fig. 8) and FFT pattern analysis (inset in Fig. 1f) showed no sign of ordering in the FCC matrix.”

Reviewer #3

The formation of deformation twins for an alloy with the stacking fault energy (SFE) of $\sim 79 \text{ mJ}\cdot\text{m}^{-2}$ is the central theme of the manuscript. If true, the result carries a very high significance because it challenges the status quo where the deformation twinning is possible only for SFE below $\sim 50 \text{ mJ}\cdot\text{m}^{-2}$.

Response: We are grateful to the reviewer for the kind support and the helpful suggestions.

For this claim, the authors have entirely relied on thermodynamic calculation for the estimation of SFE of the compositionally complex steel studied in this work. There is no discussion on the accuracy of SFE estimation following the procedure outlined. There is also no comment on the choice of thermodynamic parameters which have been taken from literature on other steels with different compositions and utilized for the present system. In literature, one can find examples of different values of SFE being reported for steels with very similar compositions using thermodynamic calculation. For example, Shun et al. [1] reported an SFE of $50.1 \text{ mJ}/\text{m}^2$ for Fe-30.9Mn-2.7Al-0.96C (wt.%) steel whereas Gutiérrez-Urrutia and Raabe [2] reported a value of $63 \text{ mJ}/\text{m}^2$ for Fe-30.5Mn-2.1Al-1.2C (wt.%) steel (Note: it is interesting to note that in ref [2] included below, Gutiérrez-Urrutia and Raabe have shown the presence of deformation twinning for Fe-30.5Mn-2.1Al-1.2C (wt.%) steel with the SFE of $63 \text{ mJ}/\text{m}^2$ which is greater than $50 \text{ mJ}/\text{m}^2$). Similarly, the choice of interfacial free energy has been taken as $9 \text{ mJ}/\text{m}^2$. Others in literature as well have used the same value for interfacial free energy for the steels with different chemical compositions (for example, Ref. [28] in the manuscript). Does it mean chemical compositions will not have any impact on the interfacial free energy?

Because of the significance of the result, the authors should have also estimated the SFE of the steel with alternative means, that is, X-ray diffraction and transmission electron microscopy to check what values of SFE the experimental techniques provided and how it compared against the thermodynamically estimated SFE.

*[1] Shun, T., Wan, C. M., & Byrne, J. G. (1992). A study of work hardening in austenitic Fe-Mn-C and Fe-Mn-Al-C alloys. *Acta metallurgica et materialia*, 40(12), 3407-3412.*

*[2] Gutiérrez-Urrutia, I., & Raabe, D. (2012). Multistage strain hardening through dislocation substructure and twinning in a high strength and ductile weight-reduced Fe-Mn-Al-C steel. *Acta Materialia*, 60(16), 5791-5802.*

Response: Thanks for the insightful comments. We organize the response into three sections:

(1) Thermodynamic parameters used for SFE estimation

Thermodynamic parameters relating to Fe, Mn, Al, and C were taken from Yoo’s work with the composition of Fe–28Mn–9Al–0.8C, wt. % (J.D. Yoo and K.-T. Park, *Mater. Sci. Eng. A*, 496, 417–424, 2008), which has a very similar alloying content of Mn, Al, and C to our steel. The additional thermodynamic parameters relating to Ni were taken from the well-established Fe-Cr-Ni alloy system (S. Curtze *et al.*, *Acta Mater.*, 59, 1068–1076, 2011).

Regarding the interfacial free energies, a value of $\sim 10 \text{ mJ/m}^2$ is used for the Fe-Mn-Al-C steel system, as shown in Table R2 below. For the Fe-Cr-Ni alloy system, a value slightly lower than 10 mJ/m^2 is typically applied for the calculation, which suggests that the interfacial free energy is not independent of the chemical compositions. But overall, the interfacial energies only show very limited change with the composition variation in steels.

Table R2 Values of interfacial free energies used in literature

Compositions (wt. %)	Interfacial free energy (mJ/m^2)	References
Fe-30Mn-3Al-3Si	10 ± 5	S. Vercammen, et al. , Steel Res. , 74 (6), 370–375, 2003.
Fe-22Mn-0.6C	10 ± 5	S. Allain, et al. , Mater. Sci. Eng., A , 387–389, 158–162, 2004.
Fe-25Mn-3Al-3Si	10	O. Grässel, et al. , J. Phys. IV France , 7, 383–388, 1997.
Fe-Mn system	15	Y.K. Lee and C.S. Choi, Metall. Mater. Trans. A , 31A, 355–360, 2000.
Fe-Cr-Ni system	7	P.J. Ferreira and P. Müllner, Acta Mater. , 46, 4479–4484, 1998.
Fe-Mn-Al-C system	10 ± 5	W.S. Yang and C.M. Wan, J. Mater. Sci. , 25, 1821–1823, 1990.

Modification: On page 20, we added several sentences to clearly point out the choice of thermodynamic parameters and discuss the influence of chemical composition on interfacial free energy: “Thermodynamic parameters for Fe, Mn, Al, and C were taken from ref.²⁸ for the Fe-Mn-Al-C alloy system, and those for Ni were obtained from the Fe-Cr-Ni alloy system²⁶. It should be noted that the value of the interfacial energy is not sensitive to compositional variations of Fe-Mn-Al-C steels⁴².”

(2) Validation of the SFE calculation by experimental results

Following the reviewer's suggestions, we performed both XRD and TEM experiments to measure the SFE of our steel, so as to validate the calculation results obtained based on thermodynamic modeling.

(i) Measurement of SFE based on XRD

The SFE (Γ) of alloys can be estimated by using the following equation (R. Schramm and R. Reed, Metal. Trans. A, 6, 1345-1351, 1975):

$$\Gamma = \frac{K_{111}\omega_0 G_{111} a_0 A^{-0.37}}{\pi\sqrt{3}} \cdot \frac{\langle \varepsilon_{50}^2 \rangle_{111}}{\alpha}$$

where $K_{111}\omega_0$ is a proportionality constant, G_{111} is the shear modulus in the (111) fault plane, a_0 is the lattice parameter, A is Zener anisotropy, ε is the micro-strain, and α is stacking-fault probability. The stacking-fault probability (α) was measured by comparing the relative peak shift between neighboring peaks of (111) and (200) in the annealed (AN) and deformed (DF) specimens, which can be calculated by (R. Schramm and R. Reed, Metal. Trans. A, 6, 1345-1351, 1975):

$$\Delta 2\theta = (2\theta_{200} - 2\theta_{111})_{DF} - (2\theta_{200} - 2\theta_{111})_{AN} = -\frac{45\sqrt{3}}{\pi^2} [\tan\theta_{200} + \frac{1}{2}\tan\theta_{111}]_{AN}\alpha$$

The 2θ values of (111) and (200) diffraction peaks of our steel are given in the following Table R3 based on XRD analysis. However, we obtained a negative value for the stacking-fault probability (α), after putting the numbers (Table R3) into the equation. This is due to the existence of $L'1_2$ structured κ -carbide (ordered FCC) in our steel, of which the peaks are overlapped with the FCC matrix based on normal XRD analysis. The κ -carbides were cut through by dislocations upon deformation, leading to the dissolution of precipitates (M.J. Yao *et al.*, Acta Mater., 140, 258-273, 2017). Thus, this unique microstructural evolution accounts for the abnormal shift of the diffraction peaks after plastic deformation.

Table R3 The 2θ values of (111) and (200) diffraction peaks

Peaks	Annealed	Deformed
111	42.360	42.541
200	49.340	49.592

The diffraction peaks of the κ -carbides and of the FCC matrix can be separated when synchrotron XRD is employed. Yet, the satellite peaks from the κ -carbides surround the FCC matrix peaks (see Fig. R9 below), which makes it difficult to measure the X-ray diffraction line profile broadening.

As a result, reliable measurement of micro-strain and further SFE is almost impossible, even though synchrotron XRD has been used.

Fig. R9 Synchrotron x-ray diffraction spectrum of the aged Fe-30Mn-8Al-1.2C (wt.%) steel (M. J. Yao, PhD thesis, RWTH Aachen University, Aachen, 2017)

Therefore, we are unable to obtain a reliable measurement of SFE based on XRD analysis in our steel with the existence of ordered FCC structured κ -carbide.

(ii) Measurement of SFE based on TEM

The SFE (Γ) also can be calculated from the Shockley partial separation distance (d) at the beginning of plastic deformation based on the following equation (J. Kim *et al.*, *Scr. Mater.*, 65, 363–366, 2011):

$$\Gamma = \frac{Gb_p}{8\pi d} \frac{2 - \nu}{1 - \nu} \left(1 - \frac{2\nu \cos 2\beta}{2 - \nu} \right)$$

where G is the shear modulus, b_p is the magnitude of the Burgers vector of the partial dislocations, ν is the Poisson's ratio, and β is the dislocation character angle (angle between the Burgers vector of the perfect dislocation and the dislocation line).

We next attempted to detect the dissociation of dislocations by performing LAADF-STEM analysis under the two-beam condition for our steel at a strain of 2% (see Fig. R10 below). The STEM image indicates that the initial deformation of our steel is controlled by full dislocations,

and no partial dislocations can be seen. This is in good agreement with our SFE calculation. As the SFE of our steel is very high, the stress is unable to cause the dissociation of dislocations at the early stage of deformation.

Fig. R10 Deformation microstructures of our steel at a strain of 2 % based on LAADF-STEM analysis. The inserted zoom-in image shows the slip of full dislocations.

Therefore, we are unable to obtain a reliable measurement of the SFE based on TEM analysis as well, but the TEM observations support the very high SFE of our steel.

(iii) Validation of our SFE calculation by experimental results from the literature

In order to further validate the estimation of the SFE based on the thermodynamic modeling approach, we also compare the calculated results by our model with the experimental results from the literature in various Fe-Mn-Al-C steels, as shown in Table R4 below. Different methods, including XRD, TEM, and neutron diffraction, have been applied to measure SFEs of steels, all of which produce values close to the calculated ones based on our model. This firmly proves the reliability of our calculation of the SFE.

Table R4 Comparison of calculated SFEs by our model with experimental results from the literature

Compositions (wt. %)	Experimental methods	Experimental SFE (mJ/m ²)	Calculated SFE by current model (mJ/m ²)
Fe-18Mn-0.6C	TEM	13 ± 3 [1]	14
	XRD	17 ± 3 [2]	
	Neutron diffraction	21 ± 4 [3]	
	XRD	19 ± 3 [4]	
Fe-18Mn-1Al-0.6C	XRD	30 ± 2 [2]	20
Fe-18Mn-1.5Al-0.6C	TEM	30 ± 10 [1]	24
	Neutron diffraction	29 ± 3 [3]	
Fe-18Mn-2Al-0.6C	XRD	36 ± 2 [2]	27
Fe-18Mn-3Al-0.6C	Neutron diffraction	44 ± 5 [3]	33

References:

- [1] J. Kim, S.-J. Lee, B.C. De Cooman, Effect of Al on the stacking fault energy of Fe–18Mn–0.6C twinning-induced plasticity, *Scr. Mater.*, 65 (2011) 363-366.
- [2] J.E. Jin, Y.K. Lee, Effects of Al on microstructure and tensile properties of C-bearing high Mn TWIP steel, *Acta Mater.*, 60 (2012) 1680-1688.
- [3] J.S. Jeong, W. Woo, K.H. Oh, S.K. Kwon, Y.M. Koo, In situ neutron diffraction study of the microstructure and tensile deformation behavior in Al-added high manganese austenitic steels, *Acta Mater.*, 60 (2012) 2290-2299.
- [4] K. Jeong, J.-E. Jin, Y.-S. Jung, S. Kang, Y.-K. Lee, The effects of Si on the mechanical twinning and strain hardening of Fe–18Mn–0.6C twinning-induced plasticity steel, *Acta Mater.*, 61 (2013) 3399-3410.

In summary, the TEM observations prove the very high SFE of our steel although it cannot give the exact value. The further comparison of calculated SFEs by our model with experimental results from literature in different kinds of Fe-Mn-C steel clearly validates the reliability of our thermodynamic modeling approach.

Modification: We added a new LAADF-STEM image in Extended Data Fig. 2 (page 26). We also added the comparison of the calculated SFEs by our model with experimental results from the literature in the Extended Data Table S2 (page24). In addition, we added two sentences in the manuscript to discuss the validation of the SFE calculation on page 20: “Such high SFE is also supported by the zoom-in LAADF-STEM observations at a strain of 2% in Extended Data Fig. 2, where no dissociation of dislocations was detected. Moreover, the thermodynamic model used here is validated by SFEs measured by various methods, as reported in the literature, including X-

ray diffraction (XRD), TEM and Neutron diffraction, for several Fe-Mn-Al-C steels (see Extended Data Table 2).”

(3) The SFE of Fe-30.9Mn-2.7Al-0.96C (wt.%) and Fe-30.5Mn-2.1Al-1.2C (wt.%) steels

As pointed out by the reviewer, Fe-30.9Mn-2.7Al-0.96C (T. Shun, *et al.*, *Acta Metal. Mater.*, 40, 3407-3412, 1992) and Fe-30.5Mn-2.1Al-1.2C (I. Gutierrez-Urrutia’s al., *Acta Mat.*, 60, 5791–5802, 2012) steels have very similar composition, yet the previously reported SFEs are quite different, e.g., 50.1 mJ/m² vs. 63 mJ/m². We also calculated the SFEs of these two steels based on our model, as shown in Table R5 below for comparison. For the Fe-30.9Mn-2.7Al-0.96C steel, both our model and the literature give very close values, which supports the accuracy of our model.

For the Fe-30.5Mn-2.1Al-1.2C steel, our model gave a lower value of 48 mJ/m². Gutierrez-Urrutia’s paper did not show the calculation process and the values of parameters used to estimate the SFE. Hence, we are unable to compare the calculation procedures. As Fe-30.9Mn-2.7Al-0.96C and Fe-30.5Mn-2.1Al-1.2C steels have very similar compositions, they should also have very close SFE in principle. Both our model and literature indicate that the SFE of Fe-30.9Mn-2.7Al-0.96C steel is around 50 mJ/m². Moreover, our model is validated by both calculation results and experimental results from the literature (see Table R4). These suggest that the SFE of ~50 mJ/m² is reasonable for this type of steels.

Table R5 Comparison of SFEs from literature with calculation from our model for two Fe-Mn-Al-C steels

Compositions (wt. %)	SFEs from literature (mJ/m ²)	Calculation of SFE by current model (mJ/m ²)
Fe-30.9Mn-2.7Al-0.96C	50.1	46
Fe-30.5Mn-2.1Al-1.2C	63	48

Furthermore, the literature did not intend to differentiate these two steels regarding to the SFE, as Shun’s paper was recognized and cited in Gutierrez-Urrutia’s paper. On Page 5798 of Gutierrez-Urrutia’s paper, “Using thermodynamical data available for the FeMnAlC system [3,9,14,50,52], we estimate a SFE of 63 mJ/m² for the Fe-30.5Mn-2.1Al-1.2C (wt.%) alloy. This value is close to the reported SFE of 50 mJ/m² in a Fe-31Mn-2.7Al-1.0C (wt.%) alloy [21].” All this evidence indicates that the SFE of Fe-30.5Mn-2.1Al-1.2C steel is at the upper limit for the TWIP effect (~50 mJ/m²).

Therefore, these two steels noted by the reviewer still belong to typical TWIP steels, of which SFEs are at the upper limit of the previous TWIP effect (around 50 mJ/m²). The two papers noted by the reviewer did not diminish the novelty of our work, while instead, they further prove the significance of deformation twinning in compositionally complex steels with very high SFE.

We are grateful that the comments from the reviewer reminded us that the deviation of SFE values existed depending on parameters used in models. Thus, in the revised manuscript, we added an approximation character in front of the SFE values to avoid any possible misunderstanding.

Modification: On page 1, the character of “~” was added in front of SFE range. On page 3, “> 50 mJ/m²” is replaced by “the upper limit for twinning is ~50 mJ/m²”.

REVIEWER COMMENTS

Reviewer #2 (Remarks to the Author):

In the revision, the current work clearly highlight the impact of high-stress twinning on materials mechanical performance. The results are new and the article is well composed. Some several points must be confirmed before publication.

1. Actually, strengthening mechanism by twinning even at high stress seems not completely new, compared with common deformation twinning. I would suggest that the scientific story can go more toward toughening effect.

2. Based the response to reviewers, when talking about strengthening or strain hardening, this work is consistent with the claim by Lai et al. that deformation twin can be enabled at high stress if fracture is prolonged to occur (Z.H. Lai et al., Acta Mat., 210, 116814, 2021) .

3. Why can this steel survive under a high stress and wait for occurrence of deformation twinning? Does it originate from high entropy effect or compositional complexity? or, Can it be explained by evolution of dislocation density due to complex strengthening effects?

4. Therefore, I suggest the authors to revise the abstract for the scientific novelty 1) high stress twinning enables materials toughening and 2) nano/microstructural complexity enables high stress twinning. The abstract now emphasize too much on the difficulty of high stress twinning. In this short letter, it would be better shift the focus on the critical points.

Response to Reviewers' Comments

NCOMMS-21-36513A-Z

We would like to begin by thanking the reviewer for the further valuable suggestions and comments. Our response is structured as follows: The new comments from the reviewer #2 are copied below (black, italic font). For each comment, we present a response and the corresponding manuscript modifications (blue font). The amended manuscript is enclosed. The changes therein are shown in red font.

Reviewer #2

In the revision, the current work clearly highlight the impact of high-stress twinning on materials mechanical performance. The results are new and the article is well composed. Some several points must be confirmed before publication.

Response: We are grateful to the reviewer for the strong support and the further comments. We have addressed all comments and revised the manuscript.

1. Actually, strengthening mechanism by twinning even at high stress seems not completely new, compared with common deformation twinning. I would suggest that the scientific story can go more toward toughening effect.

Response: Thanks for the useful comments. Indeed, the toughening effect from the deformation twinning provides additional benefits to mechanical performance. We have added a sentence in the discussion section to further clarify the importance of an additional toughening effect by high stress twinning.

Modifications: On pages 7-8, we have revised the manuscript to emphasize the toughening effect by deformation twinning: “In addition, the formation of nanotwins can impede the propagation of cracks, thus contributing a significant toughening effect^{31,34}. All of these mechanisms delay the nucleation and propagation of cracks...”

2. *Based the response to reviewers, when talking about strengthening or strain hardening, this work is consistent with the claim by Lai et al. that deformation twin can be enabled at high stress if fracture is prolonged to occur (Z.H. Lai et al., Acta Mat., 210, 116814, 2021).*

Response: We agree with the reviewer. The observed trend on strengthening and strain hardening offered by deformation twinning in the current study is consistent to the findings in Lai *et al.*'s paper. We have cited Lai *et al.*'s paper (Acta Mater., 210, 116814, 2021) in the revised manuscript.

Modifications: On page 7, we have revised the manuscript and referred to Lai *et al.*'s paper: “The formation of nanotwins in turn enables further strain hardening and toughening reserves during the later stages of deformation, which counteract softening and strain localization, as seen in some HEAs^{19,30,31} and lightweight steels^{32,33}.”

3. *Why can this steel survive under a high stress and wait for occurrence of deformation twinning? Does it originate from high entropy effect or compositional complexity? or, Can it be explained by evolution of dislocation density due to complex strengthening effects?*

Response: We appreciate the reviewer's comments. Yes, the dual-nanoprecipitation behavior, which has been achieved due to our unique design strategy based on the compositionally flexible concept of high-entropy alloys, shifts the flow strength into a regime where mechanical twinning can be activated prior to damage initiation. During the deformation, the complex strengthening and strain hardening effects, resulting from the increasing of dislocation density and the associated interactions between dislocations and nanoprecipitates, play critical roles in reaching the critical stress for twinning. We have further revised the manuscript to clearly illustrate these aspects of our study.

Modifications: We have updated the sentences on page 7: “We adopt the HEA concept to redesign lightweight steels by enhancing their compositional complexity, achieving the previously unattainable dual-nanoprecipitation system in our CCS...which arises from the increasing dislocation densities and the associated complex interactions between dislocations and nanoprecipitates (Supplementary Fig. 2)...”

4. Therefore, I suggest the authors to revise the abstract for the scientific novelty 1) high stress twinning enables materials toughening and 2) nano/microstructural complexity enables high stress twinning. The abstract now emphasize too much on the difficulty of high stress twinning. In this short letter, it would be better shift the focus on the critical points.

Response: Thanks for the valuable comments. We have shortened the descriptions on the difficulty of twinning in high stacking fault energy materials. The revised abstract now reveals the scientific novelties of toughening from the high stress twinning and nano/microstructural complexity.

Modifications: On page 1, we have updated the abstract following the reviewer's suggestions.

REVIEWERS' COMMENTS

Reviewer #2 (Remarks to the Author):

I appreciate for the revision from the authors. I have no major comments on this work and recommend for its publication in Nature Communications.

Two minor suggestions:

1. Line 30: Plastic mode should include phase transformation.
2. Twinning stress calculation: Can the authors provide a range of twinning stress. The partially recrystallized microstructure will give you a Taylor factor deviated from 3.06.

Response to the Reviewer's Comments

Reviewer #2

I appreciate for the revision from the authors. I have no major comments on this work and recommend for its publication in Nature Communications.

Response: Thanks for the suggestions. We have addressed the two comments and revised the manuscript, as detailed below.

Two minor suggestions:

1. Line 30: Plastic mode should include phase transformation.

Response: The “displacive phase transformation” has been added in the sentence as one of the plastic deformation modes.

Modifications: On page 2, we have revised the sentence: “The plastic deformation mechanisms that govern the mechanical performance of crystalline metallic materials include dislocations, twinning, stacking faults, and displacive phase transformations¹.”

2. Twinning stress calculation: Can the authors provide a range of twinning stress. The partially recrystallized microstructure will give you a Taylor factor deviated from 3.06.

Response: Indeed, the microstructure, specifically the kinematics of the deformation-induced grain rotations, can influence the Taylor factor. To address this, we have also referred to the crystal plasticity finite element model that considers the realistic deformation behavior of each grain, which gives the value of 2.7 for the Taylor factor. This value is slightly lower than the value of 3.06 from the analytical Taylor model. Based on the corresponding analysis, we have provided a range of twinning stress, i.e., 1.5–1.7 GPa.

Modifications: We have added the range of twinning stress on pages 6 and 12. On page 11, we have added a sentence to note the variation of Taylor factor: “ $M=3.06$ based on the Taylor model and $M=2.7$ based on the crystal plasticity finite element model³⁹.”